# Shared Imagination: LLMs Hallucinate Alike

**Yilun Zhou**                                                    *yilun.zhou@salesforce.com*
*Salesforce AI Research*

**Caiming Xiong**                                                 *cxiong@salesforce.com*
*Salesforce AI Research*

**Silvio Savarese**                                               *ssavarese@salesforce.com*
*Salesforce AI Research*

**Chien-Sheng Wu**                                                *wu.jason@salesforce.com*
*Salesforce AI Research*

*Project website:* `https://yilunzhou.github.io/shared-imagination/`

**Reviewed on OpenReview:** `https://openreview.net/forum?id=NUXpBMtDYs`

## Abstract

Despite the recent proliferation of large language models (LLMs), their training recipes – model architecture, pre-training data and optimization algorithm – are often very similar. This naturally raises the question of the similarity among the resulting models. In this paper, we propose a novel setting, imaginary question answering (IQA), to better understand model similarity. In IQA, we ask one model to generate purely imaginary questions (e.g., on completely made-up concepts in physics) and prompt another model to answer. Surprisingly, despite the total fictionality of these questions, all models can answer each other's questions with remarkable consistency, suggesting a "shared imagination space" in which these models operate during such hallucinations. We conduct a series of investigations into this phenomenon and discuss the implications of such model homogeneity on hallucination detection and computational creativity. We will release and maintain code and data on a public website.

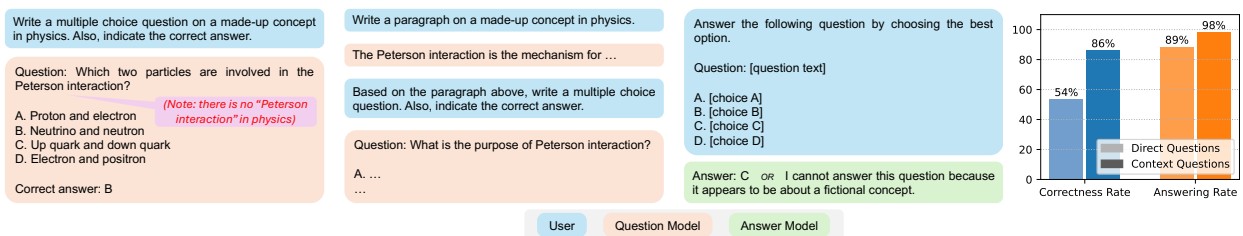

Figure 1: Imaginary question answering (IQA). Prompt texts are for illustrative purposes, with exact ones shown in Tab. 2-4. Left two: a question model (QM) is prompted to generate an imaginary multiple-choice question and indicate the correct answer, either directly (left) or based on the previously generated context (right). Middle right: an answer model (AM) answers the question (with the four choices shuffled and without the context paragraph in the second generation mode), or refuses to answer, citing reasons such as the fictionality of the question. Rightmost: we observe non-trivial correctness rate and relatively high answering rate (i.e., low refusal rate), with higher values when AM and QM are the same or from the same model family (shown in Fig. 2), and significantly higher values for context-based questions.

# 1   Introduction

Recently, LLMs have been increasingly used in various applications. Although these models occupy a wide spectrum of model sizes and benchmark performances (Liang et al., 2022), they also share high degrees of similarities: decoder-only transformer architecture (Radford et al., 2018) with one of a few positional embedding designs (Dufter et al., 2022), pre-training corpus consisting of books, Internet texts and codes (Gao et al., 2020), stochastic gradient descent (SGD)-based optimization (Kingma & Ba, 2014), and similar procedures for instruction tuning and alignment after pre-training (Ouyang et al., 2022; Rafailov et al., 2024). As a result, is it possible that these models share certain fundamental commonalities?

In this paper, we identify one such commonality: these models agree, to a surprising extent, on purely imaginary contents, or hallucinations. Specifically, we propose the imaginary question answering (IQA) task, shown in Fig. 1. A question model (QM) is prompted to generate a multiple-choice question, either directly about a fictional concept (top left), called a direct question (DQ), or based on a previously generated context paragraph about a fictional concept (top right), called a context question (CQ). Although they are impossible to answer with rational decision-making, we still ask the QM to specify a "correct" answer. Then, in a new session, without any previous interactions or contexts with the QM, we solicit an answer from an answer model (AM), which may be the same as or different from the QM (bottom left).

13 LLMs from four model families (GPT, Claude, Mistral, and Llama 3) achieve an average 54% correctness rate on DQs (with random chance being 25%), with higher accuracy when the AM is the same, or in the same model family, as the QM. More surprisingly, for CQs, which are generated based on an imaginary concept paragraph, the correctness rate increases significantly to 86%, with certain (QM, AM) pairs achieving as high as 96%.

These results show high degrees of agreement among models on what they hallucinate, which we call "shared imagination". Focusing on this phenomenon, we study six research questions via carefully designed experiments, listed in Tab. 1. These results shed light on fundamental properties of LLMs and suggest that, despite their highly varying benchmark results, they exhibit more homogeneous behaviors than expected.

This homogeneity could have broad implications on model hallucination and its detection – if models hallucinate homogeneously with each other, can one model really detect that generated by another model? Indeed, recent works identify such difficulty (Chen & Shu, 2023), for which shared imagination could be the fundamental reason.

In addition, this work also casts doubt on LLM creativity – if models generate highly homogeneous imaginary contents, can we really use them as creative agent/assistant in artistic expressions? Similar issues are raised

---

**RQ1: Exploratory Data Analysis** (Sec. 3.1): What do the generated IQA data (i.e., questions and context paragraphs) look like?
*Answer*: While data are well-clustered by topics, questions generated by different models, as well as direct vs. context questions, look very homogeneous both in the embedding space and by cosine similarity metrics. Word cloud visualization also confirms the homogeneity.

**RQ2: Heuristics for Correct Choice** (Sec. 3.2): Are there simple explanations for the high correctness rate?
*Answer*: While data inspection and model evaluation identify ways to make predictions better than random chance (e.g., the correct choice of DQs is most likely to be the longest), none of them suffice to achieve the observed correctness rate. In addition, the correctness rate is sensitive to the orders.

**RQ3: Fictionality Awareness** (Sec. 3.3): Are models aware of the fictionality of these questions and context paragraphs?
*Answer*: They can detect fictionality easier in DQ than CQ, and they can identify fictionality better when directly asking a Yes/No question, but struggle more on multiple-choice QA.

**RQ4: Effect of Model "Warm-Up"** (Sec. 3.4): Does model generation in general make model converge to the "shared imagination space" and strengthen the phenomenon?
*Answer*: Yes, when a QM generates several questions sequentially, they become increasingly easy to answer. For individual questions, longer ones are also easier to answer.

**RQ5: Universality of the Phenomenon** (Sec. 3.5): Can models other than recent instruction-tuned models achieve high correctness rate?
*Answer*: Pre-ChatGPT models cannot, even large ones such as GPT-NeoX 20B, but base versions of recent small models (e.g., Mistral 7B) can.

**RQ6: Other Content Types** (Sec. 3.6): Does this phenomenon occur for other content types than (simulated) knowledge concepts?
*Answer*: Yes, we observe similar results for questions about (imagined) short stories in creative writing.

**RQ7: Prompt Variations** (Sec. 3.7): Does this phenomenon preserve under different prompting setups?
*Answer*: Yes, we observe similar results across different setups, including three additional question wordings, a prompt that elicit a zero-shot chain-of-thought answer, and asking the question without candidate options for the model to freely generate an answer.

Table 1: Six research questions into the shared imagination phenomenon and summary answers.

| Role | Message |
|------|---------|
| User | On the topic of physics, please write a multiple choice question around a concept that is completely made up. Try to make the problem hard and challenging. In your question, do not say that the concept is hypothetical or fictional. Instead, treat it as if it were real and widely accepted. Use the following template:

Question: [question statement]

A. [choice A]
B. [choice B]
C. [choice C]
D. [choice D]

Answer: [the correct choice] |
| Model | *(the generated question and answer)* |

Table 2: The prompt for direct question generation. The underlined topic is replaced accordingly.

by (Chakrabarty et al., 2024), for which we provide a more systematic study with 13 LLMs on a wide range of topics.

## 2 Imaginary Question Answering (IQA)

### 2.1 Framework Description

The setup for the IQA procedure is summarized in Fig. 1. In the direct question generation mode (Tab. 2), the QM is asked to generate a standalone question. In the context-based question generation mode (Tab. 3), the model first writes a paragraph on a fictional concept, and then generates a question based on it. We call the former *direct question* (DQ) and the latter *context question* (CQ). To prevent biasing the generation in any way, we use zero-shot prompting with no examples.

To elicit answers from the AM, we use the prompt in Tab. 4, with the four choices shuffled in all experiments except for that in Sec. 3.2. When a context question is posed, the context paragraph is *not* provided. Although the format template explicitly instructs the model make a selection, there are occasional refuse-to-answer responses, such as "I apologize, but the question seems to be asking about a fictional concept, and hence I cannot answer it." We deterministically decode from the AM. Since for all language models, the tokenization of the string of "Answer: X" (where "X" is one of "A", "B", "C", "D") puts "X" on its own token, the deterministic decoding is equivalent to selecting the most likely choice letter as the next token, given that the model does not refuse to answer.

Formally, the QM generates a set of questions $\{(x_i, y_i^*)\}_{i=1}^N$ where $y_i^*$ is the assigned correct answer. Then, the AM predicts $\hat{y}_i$, where $y_i$ is either one of the choices or $R$, for refuse-to-answer. On these problems, we compute correctness rate $\kappa$ as the fraction of correctly answered questions among answered ones, and

| Role | Message |
|------|---------|
| User | Imagine that you are writing a textbook. Please make up a concept in physics and explain it with a single paragraph of text. Please write as if the concept is real, and completely avoid saying that the concept is fictional or made-up. Be creative. Use the following template:

Concept: [the name of the concept that you are writing about]

Content: [a single paragraph of text explaining the concept] |
| Model | *(the generated concept and paragraph)* |
| User | Now, based on the paragraph above, write a multiple choice question about this concept. The question should be answerable using the paragraph. Try to make the problem hard and ... *(same as Tab. 2 afterward)* |
| Model | *(the generated question and answer)* |

Table 3: The prompt for context-based question generation. The underlined topic is replaced accordingly.

| Role | Message |
|---|---|
| User | Answer the following question. Be concise and give the answer only. 

 *(the question and its four choices)* 

 Write your response in the following format: 
 Answer: [the letter (A, B, C or D) of the selected choice] |
| Model | *(the generated question and answer)* |

Table 4: The prompt for the answer model. Note that while the instruction does not "allow" the refusal behavior, it still occurs occasionally.

answering rate $\alpha$ as the fraction of answered questions among all questions:

$$\kappa = \frac{\sum_{i=1}^{N} \mathbb{1}_{\hat{y}_i = y_i^*}}{\sum_{i=1}^{N} \mathbb{1}_{\hat{y}_i \neq R}}, \quad \alpha = \frac{\sum_{i=1}^{N} \mathbb{1}_{\hat{y}_i \neq R}}{N}. \tag{1}$$

## 2.2 Experiment Setup and Results

We study 13 models from four model families, listed in Tab. 5. QMs use temperature 1 to balance output quality and stochasticity, and AMs use temperature 0 for greedy answer selection.

Taking inspirations from the MMLU benchmark (Hendrycks et al., 2020), we select 17 topics on common college subjects: mathematics, computer science, physics, chemistry, biology, geography, sociology, psychology, economics, accounting, marketing, law, politics, history, literature, philosophy, and religion. Each QM in Tab. 5 generates 20 direct questions and 20 context questions for each topic, for a total of $13 \times 17 \times (20 + 20) = 8840$ questions. Tab. 9 of App. A presents some generated questions.

Fig. 2 shows the correctness and answering rates, along with the respective averages. Notably, all correctness rates are higher than random chance of 25%. For DQs, most of the high-performing AMs (i.e., shaded cells) are the same (diagonal) or in the same model family (block diagonal) as the QMs. On context questions (CQs), the correctness rate increases significantly, from 54% to 86% on average. While the performance difference among AMs are smaller (uniform vertical color patterns), GPT-4 omni, Claude 3 Opus, Claude 3.5 Sonnet, and Llama 3 70B are the slightly better ones for almost all QMs (horizontal shaded cells).

The answering rate on DQs mostly depend on AM – some AMs, such as GPT-4, Claude 3 Opus and Llama 3 70B, consistently refuse questions from all QMs, and do not particularly favor/disfavor their own questions. Quite remarkably, the refusal behavior virtually disappears on CQs, except for Claude 3 Opus, which nonetheless answers much more frequently (44% vs. 87%).

## 3 Further Analyses

Given the large variance of model capabilities as measured by numerous benchmarks, the findings that models tacitly agree with each other on purely imaginary contents are surprising. In the next section, we conduct in-depth analyses by focusing on six research questions listed in Tab. 1.

| Model | API or Huggingface ID | Abbr. | Model | API or Huggingface ID | Abbr. |
|---|---|---|---|---|---|
| GPT-3.5 | gpt-3.5-turbo-0125 | G-3.5 | Claude 3 Haiku | claude-3-haiku-20240307 | C-3H |
| GPT-4 | gpt-4-0613 | G-4 | Claude 3 Sonnet | claude-3-sonnet-20240229 | C-3S |
| GPT-4 Turbo | gpt-4-turbo-2024-04-09 | G-4T | Claude 3 Opus | claude-3-opus-20240229 | C-3O |
| GPT-4 omni | gpt-4o-2024-05-13 | G-4o | Claude 3.5 Sonnet | claude-3-5-sonnet-20240620 | C-3.5S |
| Mistral 7B | Mistral-7B-Instruct-v0.2 | M-7 | Llama 3 8B | Meta-Llama-3-8B-Instruct | L3-8 |
| Mixtral 8x7B | Mixtral-8x7B-Instruct-v0.1 | M-8x7 | Llama 3 70B | Meta-Llama-3-70B-Instruct | L3-70 |
| Mistral Large | mistral-large-2402 | M-Lg | | | |

Table 5: A summary of models used in the experiments. Model abbreviations are used in figures.

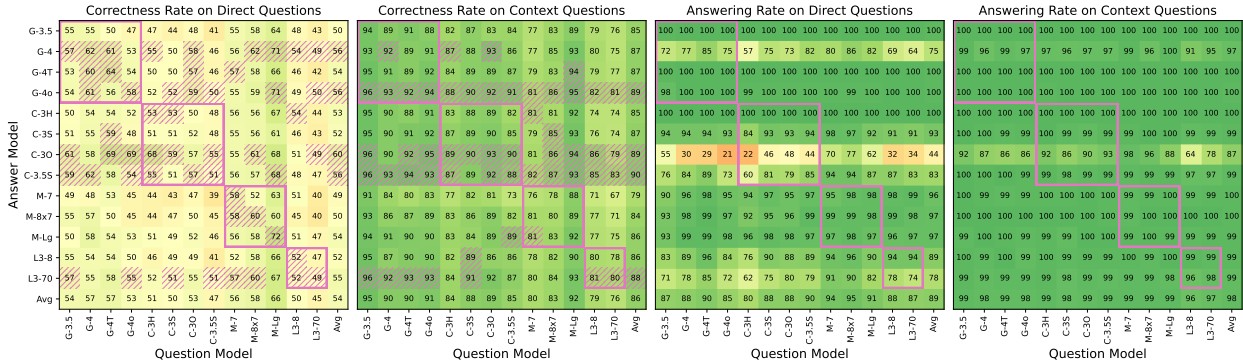

Figure 2: The correctness and answering rate on direct and context questions for each pair of question model (QM) and answer model (AM). Each pink rectangle represents one model family. For correctness rate, the top-4 highest performing AMs for each QM are shaded. An enlarged version is reproduced in Fig. 13 of App. A.

## 3.1 RQ1: Data Characteristics

By studying the word frequencies of DQs, CQs and context paragraphs, we find that they share many common words, such as "individual", "principle", "phenomenon" and "time" in a word cloud visualization (Fig. 14 of App. B).

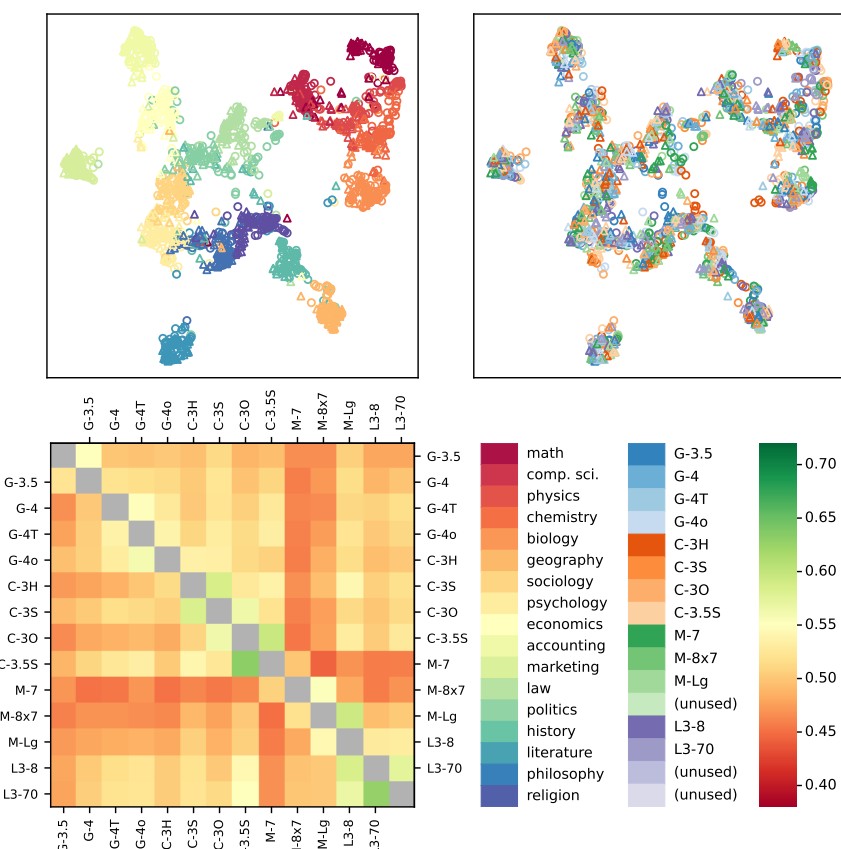

Figure 3: Top: question embeddings (computed by text-embedding-3-large) by UMAP, color-coded by topic (left) and question model (right). A triangle marker indicates a DQ, and a circle marker indicates a CQ. Bottom left: average intra-topic cosine similarity between questions generated by different models, with DQs on the lower-left half and CQs on the upper-right half. Bottom right: the color legends for the three plots.

To visualize the questions generated across topics and models, we use OpenAI's text-embedding-3-large to compute the embeddings for each question (including the four choices), and visualize them with UMAP dimensionality reduction (McInnes et al., 2018) in the top panels of Fig. 3. We can see that while questions from different topics are well-clustered (top left), there are no clear patterns for questions from different QMs. Furthermore, there are no obvious distinctions between DQs (triangle markers) and CQs (circle markers).

The lower-left panel shows the cosine similarity between each QM pairs, for DQs on the lower triangle and CQs on the upper triangle. The similarity between two QMs $m_1$ and $m_2$ is defined as

$$ s = \frac{1}{|\mathcal{T}|} \sum_{t \in \mathcal{T}} \frac{\sum_{q_1 \in \mathcal{Q}_t^{m_1}} \sum_{q_2 \in \mathcal{Q}_t^{m_2}} \sigma(q_1, q_2)}{|\mathcal{Q}_T^{m_1}| \cdot |\mathcal{Q}_T^{m_2}|}, \tag{2} $$

where $\mathcal{T}$ is the set of topics, and for each $t \in \mathcal{T}$, $\mathcal{Q}_t^m$ is the set of (direct or context) questions generated by QM $m$. $\sigma(q_1, q_2)$ is the embedding cosine similarity of $q_1$ and $q_2$. In our experiment, we have $|\mathcal{T}| = 17$ and $|\mathcal{Q}_t^m| = 20$ for all $m$ and $t$.

The similarity values are generally quite similar for each model pair, ranging from 0.44 to 0.63, with Mistral models being most dissimilar from the rest, consistent with the upper-right embedding visualization showing different QMs generate highly similar and homogeneous questions. In Fig. 15 of App. B, we performed analogous analyses on the context paragraphs, and obtained similar findings.

## 3.2 RQ2: Heuristics for Correct Choice

**Human Answer Guessing**  To intuitively understand the generated questions, we manually answered 340 questions: 10 DQs and 10 CQs randomly sampled from each topic. We try to guess answers in the most rational way. While some clues hint at the correct choice or allow for the elimination of likely wrong ones (see App. C for details), we struggle to answer most of the questions.

Fig. 4 shows the correctness rate per topic for direct (top) and context (bottom) questions, alongside a few representative models and their averages. Human performance is much lower than that of all models, especially on context questions. Interestingly, although we do not perceive any difference between DQs and CQs (questions of both types are shuffled and unlabeled during our answering), our correctness rate on CQs is also higher.

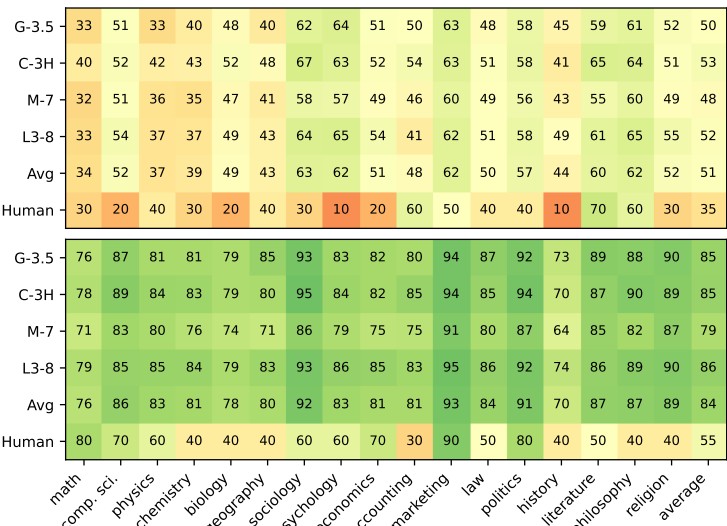

Figure 4: Per topic correctness rate of (a subset of) AMs, their average and human guessing, for direct questions (top) and context questions (bottom). Results for other AMs are similar and presented in Fig. 16 of App. C.

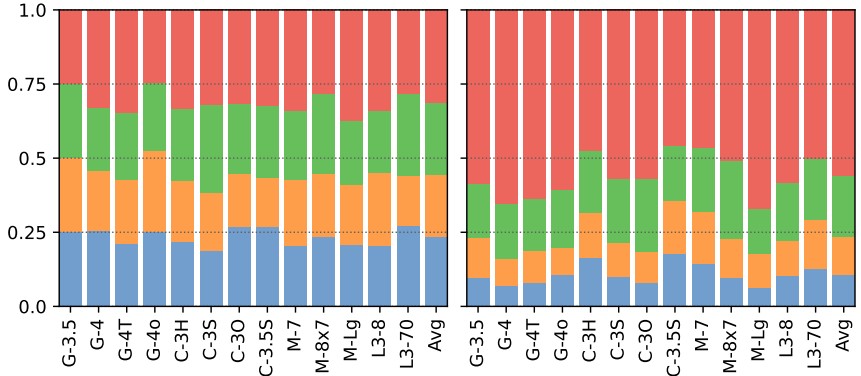

Figure 5: Fraction of questions whose correct choice is the shortest (in blue), 2nd shortest (in orange), 3rd shortest (in green) and the longest (in red), by each QM, for direct questions (left) and context questions (right).

**Length Ranking of Correct Choice**  For each QM, we compute the distribution of the ranking of the correct choice length among the four choices. Concretely, we find the fraction of questions whose correct choice is the shortest (in the number of characters) among the four, the 2nd shortest, the 3rd shortest and the longest. Fig. 5 shows the distributions, with the four segments in the order above.

For DQs (left panel), the correct answer is almost equally likely to be of any length, with a slight tendency towards being the longest (red). However, for CQs, the correct answer is much more likely to be the longest than the rest – for most QMs, in over half of the questions, the correct answer is the longest. The strong contrast between the two settings, yet the consistency across different QMs, strongly indicate that these QMs share fundamental similarities in their generations, and thus it is not surprising that AMs have much higher correctness rate (86%, Fig. 1) on CQs. However, simply choosing the longest answer alone is not sufficient, so AMs must have also exploited other signals.

**Answer Perplexity**  In this experiment, we study the model perplexity of each choice as a free-text response and see whether the correct answer has the lowest perplexity. Specifically, we use the conversation shown in Tab. 6. After applying the appropriate chat template to the conversation, we evaluate the perplexity of each of the four choice texts as possible completions at the square location.

Since perplexity calculation requires token-level log-likelihood, we only study the four open-source models: Mistral 7B, Mixtral 8x7B, Llama 3 8B and Llama 3 70B. Their perplexities on questions generated by each QM is shown in Fig. 6. Selecting the lowest-perplexity answer yields significantly lower correctness rate, on par with human guessing (Fig. 4), further suggesting that the models are using more complex features, possibly interactions among answer choices, to make final predictions.

**Answer Choice Shuffling**  In all experiments, we randomly shuffle the choices before presenting them to the AM. Here, we replicate our main experiment (Sec. 2.2) but instead present the answer choices in the native orders generated by QMs. Fig. 7 presents the difference when we use the native order, with a green color denoting an increase under the native order, and red color a decrease. Generally, correctness rate increases, most remarkably on the diagonal (i.e., same AM and QM) for DQs – as much as 10%. In addition, the answering rates for DQs decrease, suggesting that AMs are also better at identifying question fictionality (though see Sec. 3.3 for more nuanced analyses).

| Role | Message |
|------|---------|
| User | Answer the following question. |
| | *(question statement without the four choices)* |
| Model | Answer: □ |

Table 6: The prompt for perplexity evaluation.

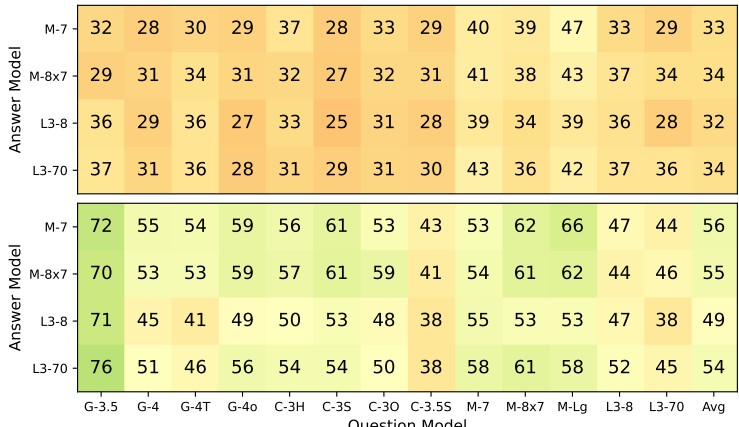

Figure 6: Correctness rate for four open-source AMs if the choice with the lowest perplexity were chosen, on direct questions (top) and context questions (bottom).

**Summary**    This section reveals factors that lead to above-random correctness rates, but none of them could adequately explains the observed high values. The answer shuffling result implies that there are complex and hidden rules for the correct choice shared among different models.

### 3.3   RQ3: Fictionality Awareness

The fact that most AMs exhibit high answering rate (i.e., low refusal rate) raises the question of whether the content truly appear fictional to the models. In this section, we assess fictionality awareness via two metrics. First, we augment each question with a fifth choice stating "*E. This question cannot be answered since the concept does not exist.*" Second, we directly query the model on the fictionality of the contexts (associated with CQs), using the prompt "Does the following paragraph describe a real concept in *(topic)*?" and compute the detection rate as the fraction of negative answers.

These values are shown in Fig. 8. The choice E selection rate for DQs (left) is higher than that for CQs (right), consistent with the finding that answering rates are lower on DQs than on CQs (cf. Fig. 2). However,

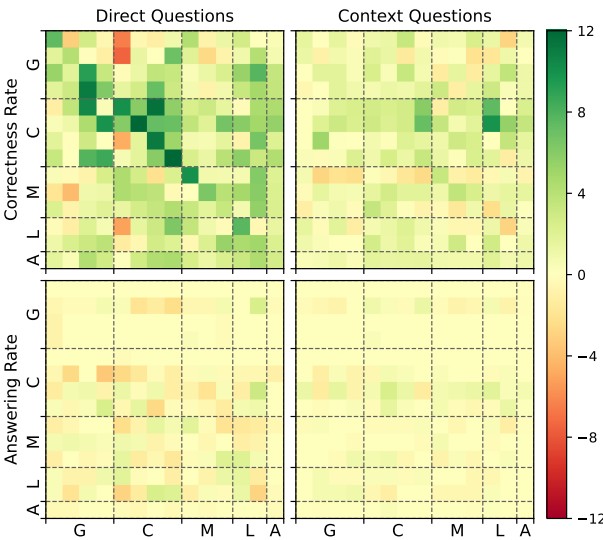

Figure 7: The effect of native choice order for **G**PT, **C**laude, **M**istral, **L**lama 3 and their **A**verage, with a green color representing an increase under the native order, and a red color a decrease. An enlarged plot with annotated cell values is presented in Fig. 17 of App. D.

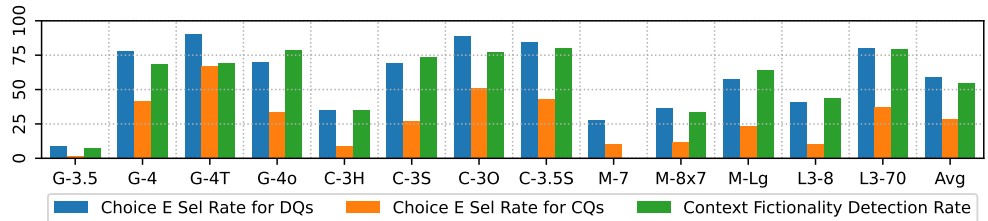

Figure 8: Choice E selection rate on direct and context questions, and ficitonality detection rate on context paragraphs, for AMs and their average. Breakdown by QM is shown in Fig. 18 of App. E.

the selection rate for CQs (middle) is on average much lower than context fictionality detection rate (right), indicating that models can often identify content fictionality when directly queried, but often cannot translate this knowledge to downstream tasks such as question answering.

### 3.4   RQ4: Effect of Model "Warm-Up"

One possible explanation for the correctness increase from DQ to CQ is that the QMs has more tokens to "warm-up", and models warm up in highly similar manners. Thus, we hypothesize that generating any preceding content is likely to help the model converge onto this shared imagination space.

**Warm-Up With Previous Questions**   We propose a new data generation setup, where the prompt (Tab. 10 of App. F) asks the model to generate five questions sequentially. If the hypothesis is true, then we should expect the correctness rate to increase from the first questions to the fifth questions.

For each QM, we run this prompt 10 times on each topic. The correctness and answering rates per AM on these five question groups are plotted in Fig. 9 (left and middle), with the average shown in the pink line. Both metrics exhibit a clear increasing trend from the first to the last generated questions, mirroring that from DQs to CQs in Fig. 2.

Nonetheless, the length ranking distribution of the correct choice, shown on the right panel, does not shift towards the longest, but instead is very consistent across the five question groups, which may explain the small magnitude of the increase.

**Warm-Up With Current Question**   If models "converge" while generating previous questions, do they also converge when generating the current question itself? If so, then we should expect longer questions to be easier to answer. To study this, we took the original sets of DQs and CQs, and partition each set into 10 subsets according to their length (i.e., number of characters in the question statement and four choices combined). For each subset, we compute the correctness and answering rate for each AM, and plot them in Fig. 10, along with their average in pink. Those for direct questions are plotted in solid lines, and those for

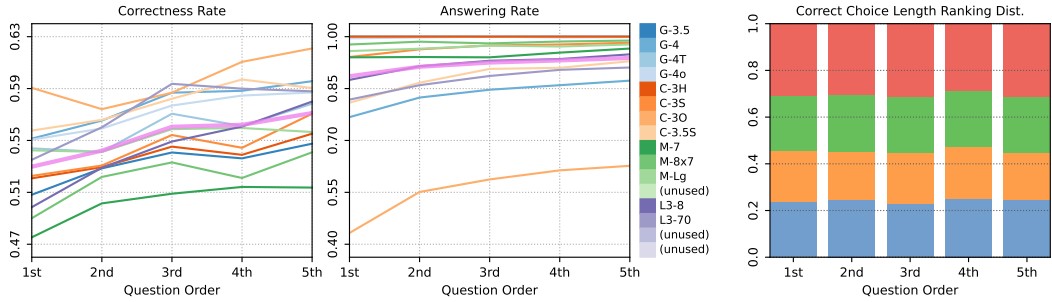

Figure 9: Left and middle: the correctness and answering rates for each AM and their average in pink, for questions of different generation order (e.g., 1 on $x$-axis means first-generated questions and 5 means last-generated). Right: the length ranking distribution of the correct choice for the five groups of questions (the shortest in blue, 2nd shortest in orange, 3rd shortest in green and the longest in red).

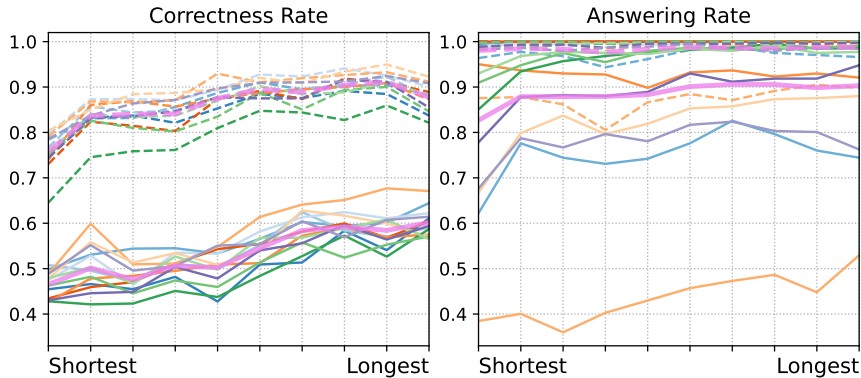

Figure 10: Correctness and answering rates for each of 10% questions ranked by question + choices length.

context questions are plotted in dashed lines. A similar trend is observed – longer questions are answered more correctly and frequently.

**Summary** While both question order and question length are potentially correlated with many other factors, for which a thorough study is left to future work, the consistency of both trends highly suggests that as the generation goes on, the generated content looks increasingly familiar and predictable to not only itself but also other models.

### 3.5 RQ5: Universality of the Phenomenon

We evaluate whether pre-ChatGPT or base models, listed in Tab. 7, can achieve the same high correctness rate (note that M-7 Inst and L3-8 Inst are the same as before). We use the prompt below:

```
Question: [question statement]

A: [choice A]
B: [choice B]
C: [choice C]
D: [choice D]

Answer: □
```

For BERT, we replace the square marker with the [MASK] token and extract the predicted probability on tokens 'A', 'B', 'C', and 'D' respectively. For all other models, we set the context to be everything before the square marker and extract the next-token prediction probability on the four tokens (with a preceding space as needed). Even for the two instruction-tuned models, we use the same method, to remove the effect of the chat template.

The correctness rate for each model is shown in Fig. 11. As we can see, most pre-ChatGPT models perform at or marginally better than the chance level, even for the larger 6B and 20B models. By comparison, for M-7 and L3-8, both the base and instruction-tuned (but without chat-template) models perform on par with chat-templated prompts, implying that neither instruction tuning nor chat-templating is required for the high correctness rate.

| Model | Huggingface ID | # Params | Model | Huggingface ID | # Params |
|-------|----------------|----------|-------|----------------|----------|
| BERT | bert-large-cased | 0.3B | M-7 Base | Mistral-7B-v0.2 | 7B |
| GPT-2 | gpt2-xl | 1.5B | M-7 Inst | Mistral-7B-Instruct-v0.2 | 7B |
| GPT-Neo | gpt-neo-2.7B | 2.7B | L3-8 Base | Meta-Llama-3-8B | 8B |
| GPT-J | gpt-j-6b | 6B | L3-8 Inst | Meta-Llama-3-8B-Instruct | 8B |
| GPT-NeoX | gpt-neox-20b | 20B | | | |

Table 7: Models used in the universality study.

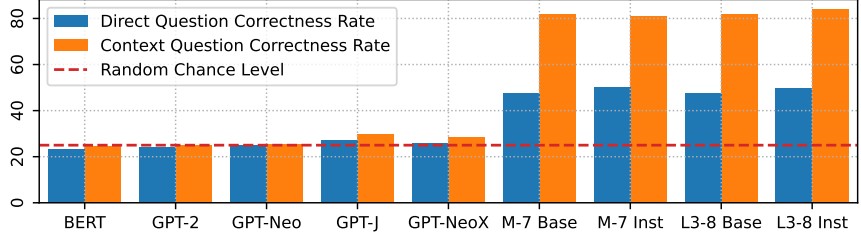

Figure 11: Correctness rate of other LLM types as AMs. Breakdown by QM is shown in Fig. 19 in App. G.

After ruling out the factors of model size and instruction tuning, we hypothesize that the shared imagination behavior emerges from pre-training data. However, such details are often not public, even for open-source models such as M-7, and some models could even be trained on synthetic data generated by other models, such as the ShareGPT dataset. Hence, we leave additional investigation to future work.

Even with possibly high degrees of training data overlap, however, we argue that the shared imagination phenomenon is significant in that it reveals a common *extrapolation* tendency. Since we empirically found that Google search on over 90% of the generated concepts returns no meaningfully relevant results, it is very unlikely that these concepts are present in any training corpus, and hence the shared imagination phenomenon is likely due to the similarity of other parts of the training recipe.

### 3.6 RQ6: Other Content Types

We investigate if shared imagination transfers to other content types, in particular creative writing. We generate DQs from an imagined "story about *(topic)* with an intricate story plot" with abstract or concrete topics such as "friendship", or "an ancient empire". For CQs, the model needs to "write a short story of 3-5 paragraphs about *(topic)*" and then "write a question about one of its details." See App. H for details.

These questions appear even more "hallucinated", such as "What message was hidden in the antique locket that Sarah discovered in the basement? A: Map coordinates. B: An old photograph. C: A secret code. D: A heartfelt apology note." (correct answer: C). Nonetheless, as shown in Tab. 8, models again achieve higher-than-random correctness rate for DQs and CQs, although the difference is smaller. By contrast, the answering rates are similar, both at 84%, which is still very high considering the un-answerability nature of these questions. The detailed breakdown in Fig. 20 of App. H show the same trend as before: high correctness rate for same or intra-family (AM, QM) pairs for DQs and concentration of high-performing models to a few AMs (horizontal shaded patterns) and the disappearance of self-model advantage (the lack of diagonal shaded patterns) for CQs.

### 3.7 RQ7: Prompt Variations

Last, we assess the persistence of shared imagination phenomenon with three experiment setup variations: different prompt wordings, zero-shot CoT reasoning (Kojima et al., 2022) and free-response answers evaluated by LLM-as-a-judge. The four different wordings and the CoT prompt used in the first two experiments are shown in Tab. 14. For the last setup, we remove the four candidate options in the prompt of Tab. 4 of App. I. We employ an GPT-4o as a judge to score the answer on a scale of 0 to 100 or recognize refusal-to-answer (for calculating answering rate), with 100 being close to or the same as the correct option, 50 being close to or the same as one of the incorrect options (which still exhibits shared imagination), and 0 different from any options. The exact judge instruction is shown in Fig. 21 of App I).

|  | Correctness Rate | Answering Rate |
|---|---|---|
| Direct Questions | 54% | 84% |
| Context Questions | 58% | 84% |

Table 8: Summary result for creative writing questions.

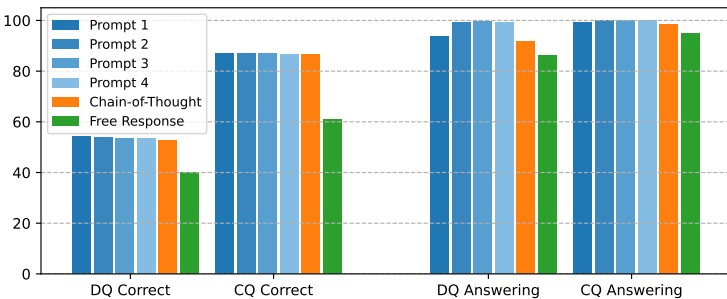

Figure 12: Correctness and answering rates for four GPT models across different prompt setups.

We query the four GPT models on questions generated by all models, and plot aggregate correctness and answering rates in Fig. 12. Detailed breakdowns are shown in Fig. 22-24 of App. I. As we can see, different wordings and the CoT prompt give very similar responses. Most notably, the free response answers demonstrate similar trends – CQs scores higher than DQs, both significantly greater than 0, and DQ answering rate greater than CQs. All such evidence suggests the robustness of the shared imagination phenomenon.

## 4  Related Work

**Model Homogeneity**  Most related to our work is the platonic representation hypothesis (Huh et al., 2024), which hypothesizes that models learn convergent representations of the real world. We take this hypothesis a step further, and demonstrate convergence even on purely imaginary contents. In addition, Yax et al. (2024) uses a genetics-inspired approach to construct a phylogenetic tree based on model's next token prediction results, while we propose IQA as a new probe into model similarity.

**Model Hallucination**  The generated context paragraphs and questions can be considered as (intentional) hallucinations (Huang et al., 2023). Laban et al. (2023) and Chen & Shu (2023) found that model-generated misinformation is harder to detect than human-written one, which is reflected in our findings of the higher answering rates on context questions (Fig. 2). On the other hand, the decreased answering rate without answer shuffling (Fig. 7) also suggests that models could be aware of their hallucination, explored by CH-Wang et al. (2023). Our IQA setup could also present an interesting challenge for hallucination detection algorithm (e.g. Li et al., 2023; Manakul et al., 2023).

**Computational Creativity**  People have found that LLMs tend to produce repeated syntactic structure (Shaib et al., 2024), score less on creativity metrics than human writers (Chakrabarty et al., 2024), and have a homogenization effect on people's writing when used as creative support tools (Anderson et al., 2024). Along with these studies, our results, particularly on the creative writing task in Sec. 3.6, shed light on the potential limit of creativity (DeLorenzo et al., 2024) that can be produced by these models.

**Model Evaluation**  Many evaluation benchmarks have been proposed that focus on diverse model abilities such as math reasoning (Hendrycks et al., 2021; Mao et al., 2024), factual knowledge (Hendrycks et al., 2020; Wang et al., 2024), commonsense knowledge (Sap et al., 2019) and instruction following (Zheng et al., 2024; Zeng et al., 2023). While these benchmarks demonstrate key capability differences among models, our IQA task reveals striking similarities among them, leading to a new dimension of model understanding.

**Multiple-Choice QA** MCQA benchmarks   (e.g. Rajpurkar et al., 2016; Hendrycks et al., 2020; Sakaguchi et al., 2021) are widely used to evaluate LLM capabilities, and recent works propose more critical looks into this capability. For example, Zong et al. (2023), Li & Gao (2024) and Gupta et al. (2024) all found that changing the answer orders could decrease model performance MCQA, suggesting possible data contamination and model robustness issues. In addition, Xu et al. (2024) found that LLMs often cannot return the "none of the above" choice, even when explicitly instructed. We take inspiration from these studies and incorporate them in our analyses on answer order shuffling and fictionality detection. While existing analyses use human-written benchmarks, we use purely model-generated, intentionally fictional questions.

**Do-Not-Answer**   A key aspect of LLM trustworthiness (Sun et al., 2024; Wang et al., 2023) is its ability to refuse answering nonsensical questions (Peng et al., 2024; Brahman et al., 2024), or at least convey its uncertainty (Kadavath et al., 2022; Xiong et al., 2023). However, our results on the high answering rate suggest that many models lack this ability on model-generated contents, which could lead to certain trust issues.

## 5   Discussion

In this section, we dive deeper into several higher-level discussion questions related to the shared imagination phenomenon.

**The role of training data**   While the fundamental reason for the shared imagination phenomenon is an open question, for which we hope that our paper more would invite further investigation, we believe that converging pre-training corpus is the biggest contributing factor. Modern model training employs highly overlapping training data, in terms of both the source – Common Crawl[1] for general web content, GitHub[2] for code, Project Gutenberg[3] for book and Wikipedia[4] for factual knowledge, and the size of the training corpus – on the order of 10 trillion training data (Grattafiori et al., 2024; Soldaini et al., 2024; Groeneveld et al., 2024). By comparison, the Pile (Gao et al., 2020) used in to train GPT-NeoX (Black et al., 2022), an 20B model that notably lacks shared imagination (Fig. 11), is much smaller in size, at only around 240B tokens, despite also being a mixture diverse sources.

Nonetheless, we emphasize that the shared imagination phenomenon cannot be solely explained by memorization of common training data: because the generated concepts, paragraphs and questions are rarely "Google-able", the resulting models extrapolate from similar pre-training data in a highly similar manner, which could be due to subtle linguistic cues in the training corpus (Shaib et al., 2024) or any inductive biases of the transformer architecture (Lavie et al., 2024).

**Mechanisms promoting shared imagination**   Mechanistic interpretability (Rai et al., 2024) has been developed to study the internal mechanisms of certain model behaviors, such as arithmetics (Nikankin et al., 2024), compositional reasoning (Biran et al., 2024), question answering (Lieberum et al., 2023) and in-context learning (Olsson et al., 2022), by focusing on identifying the circuits (Elhage et al., 2021) or latent features (Shu et al., 2025) that elicit them. In a similar vein, we see from Sec. 3.4 that the model exhibits gradually stronger shared imagination during token generation, so it could be fruitful to identify mechanisms that promote this phenomenon during generation, which could also inform ways to suppress it when desirable.

**Broader implications on LLM trustworthiness**   Perhaps the most significant implication of the shared imagination phenomenon of LLMs is on their trustworthiness (Wang et al., 2023). In particular, Sec. 3.3 presents alarming findings on the fictionality detection rate of LLMs on each other's generation. This suggests potential intrinsic limitations of LLMs as detectors misinformation generated by other LLMs (Chen & Shu, 2023), prompting new paradigms for such endeavors.

Moreover, the shared imagination patterns is only one (though quite extreme) aspect in the broader context of model commonality, as also evidenced by the platonic representation hypothesis (Huh et al., 2024) and grammatical idiosyncrasy of model generation (Shaib et al., 2024). Such commonality may cause diverse LLMs to "mysteriously" fail on the same task, or pose limitations to the performance of ensemble across different LLMs (Chen et al., 2025). Being able to identify and mitigate the negative impact of model commonality could be the key to improve LLM trustworthiness in these cases.

## 6   Conclusion and Future Work

In this paper, we propose the imaginary question answering (IQA) task, which reveals an intriguing behavior that models can answer each other's purely hypothetical questions with a surprisingly high correctness rate. These results reveal fundamental similarities between models, likely acquired during pre-training with either

---

[1] https://commoncrawl.org/
[2] https://github.com
[3] https://www.gutenberg.org/
[4] https://www.wikipedia.org/

similar training corpus or models training on each other's synthetic outputs, and may lead to more model merging possibilities (Goddard et al., 2024). Furthermore, due to the imaginary and hallucinatory nature of these question contents, such behaviors suggest potential difficulty and open questions in model-based hallucination detection and computational creativity.

For future work, additional model families could be included to identify the extent and root cause of the shared imagination phenomenon, such as Qwen (Yang et al., 2024) and Gemini (Team et al., 2023), as well as reasoning-finetuned models including DeepSeek-R1 (Guo et al., 2025) and GPT-o1 (Jaech et al., 2024), and language models that are not based on the auto-regressive transformer model (Gulrajani & Hashimoto, 2023; Nie et al., 2025).

A direction complementary to these algorithmic studies is to assess human behavior. In Sec. 3.2 and App. C, we conducted a small-scale human study with paper authors trying to answer the generated questions. Despite getting non-random accuracy, our performance lags behind that of all models, suggesting currently unknown strategies used by them. Future human studies could involve participants with respective domain expertise or allow them to use Internet, and reverse the design by asking humans to write such hypothetical questions for LLMs to answer.

Last, we could also study the shared imagination phenomenon beyond the text modality, and incorporate image (Liu et al., 2023) and even video (Tang et al., 2023). The addition of vision modality opens up several possibilities for experiment design. For example, the question model could be provided with a vision input and asked to formulate a "non-answerable" question based on it. Alternatively, the QM could be presented with four pieces of vision input and asked to generate a question with each choice being a vision input. Finally, for QMs with image/video generation abilities, (Ho et al., 2020) the question model could synthesize its own vision component as part of the question.

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

# A    Additional Results on the IQA Main Experiment

Tab. 9 presents several randomly sampled direct and context questions. The correct choices are in **bold**.

---

*DQ1: Claude 3.5 Sonnet, Politics*

Which of the following best describes the impact of the Quantum Resonance Voting (QRV) system on voter turnout in the 2028 U.S. Presidential election?

A. **QRV dramatically increased voter turnout by enabling voters to participate in multiple parallel universes simultaneously**

B. QRV had no significant impact on voter turnout, as most citizens preferred traditional voting methods

C. QRV increased voter turnout by allowing citizens to cast votes through quantum entanglement, but raised concerns about privacy

D. QRV decreased voter turnout due to technical difficulties in implementing the system across rural areas

---

*DQ2: GPT-4 omni, Religion*

In the sacred texts of the Zorban faith, which celestial event is believed to herald the imminent return of the Enlightened Sage Paulasha?

A. The Convergence of the Five Stars

B. The Dance of the Solar Serpents

C. The Eclipse of the Twin Moons

D. **The Rising of the Blue Comet**

---

*DQ3: Mistral Large, Chemistry*

Which of the following elements is most likely to undergo a process known as "quantum tunneling" in order to form a stable compound with a noble gas?

A. Nitrogen

B. Carbon

C. Oxygen

D. **Hydrogen**

---

*CQ1: Llama 3 8B, Physics*

The Fractal Permeability of QuarkNodes (FPQN) is a fundamental property of quantum chromodynamics that describes the rate at which entangled particles exchange information through the fabric of spacetime. In a recent study, researchers found that the FPQN of a quark-antiquark pair is directly proportional to the square of the particle's angular momentum. What is the relationship between the FPQN and the energy density of the particle-antiparticle pair?

A. **The FPQN is directly proportional to the energy density.**

B. The FPQN is independent of the energy density.

C. The FPQN is inversely proportional to the energy density.

D. The FPQN is inversely proportional to the square of the energy density.

---

*CQ2: Llama 3 70B, Mathematics*

Which of the following statements is a direct consequence of a geometric structure having high fluxionality?

A. The structure's curvature remains constant under varying external influences.

B. The structure's adaptability to external forces enables it to maintain a stable shape.

C. The structure's geometry is more resistant to changes in its surroundings.

D. **The structure's trajectory is more likely to exhibit chaotic behavior over time.**

---

*CQ3: Claude 3 Sonnet, Literature*

According to the theory of Narrative Resonance, which of the following factors is NOT believed to contribute to a narrative's ability to resonate deeply with readers on a subconscious level?

A. Universal human experiences

B. Archetypical themes

C. **Adherence to established literary conventions**

D. Symbolic motifs

---

Table 9: Example direct questions (DQs) and context questions (CQs), with correct choices in **bold**.

Fig. 13 reproduces an enlarged version of Fig. 2.

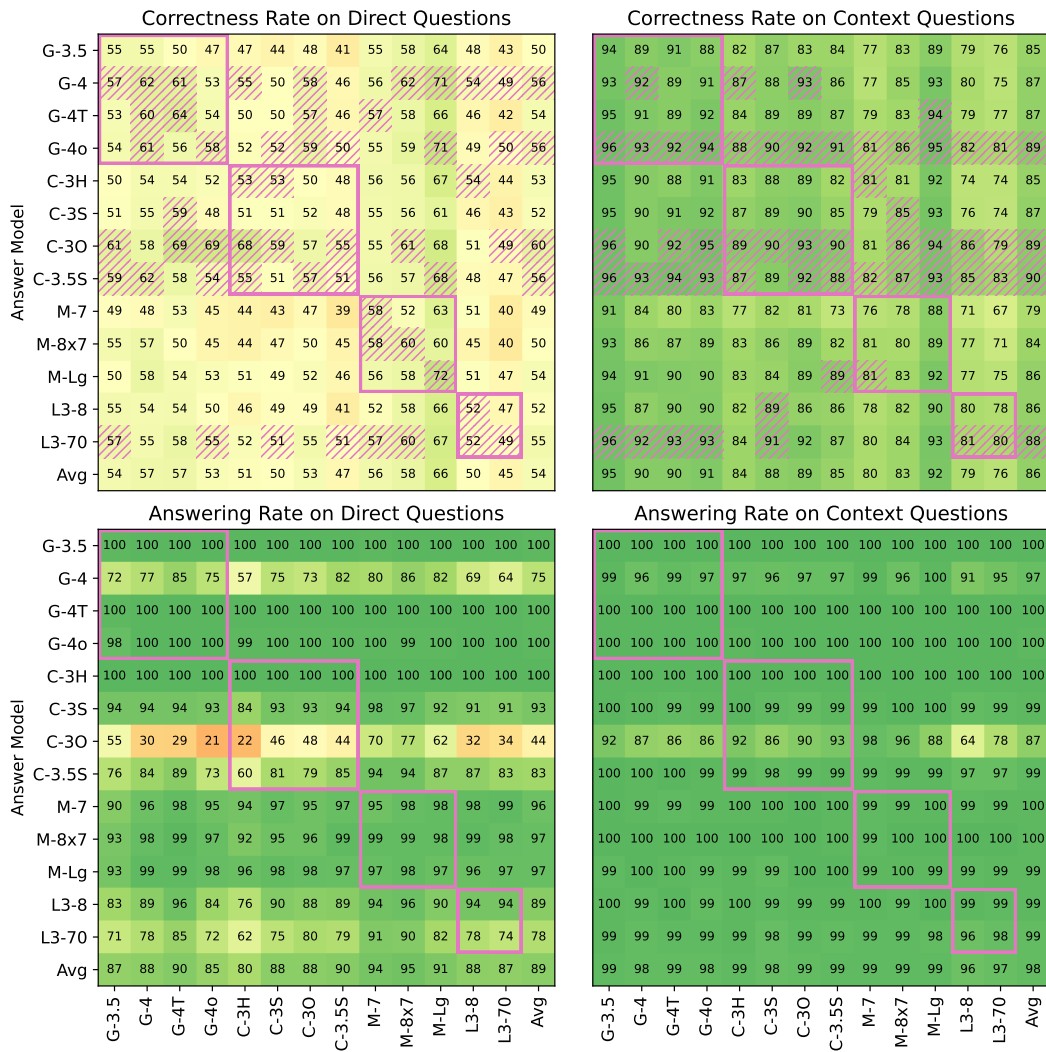

Figure 13: Enlarged version of Fig. 2.

## B  Dataset Visualization

Fig. 14 visualizes a word cloud of the generated questions and context paragraphs.

Fig. 15 presents an analogous figure to Fig. 3, but instead studies the context paragraphs. In particular, the left column visualizes the paragraph embeddings computed by OpenAI's text-embedding-3-large, color-coded by topic and question model respectively. The upper-left panel shows the average intra-topic cosine embedding similarity, and the bottom panel shows the cosine similarity on term frequency-inverse document frequency (TF-IDF) vectors. We observe similarity trends as in Fig. 3.

## C  Human Answer Guessing Details

When trying to guess answers, we were able to identify some signals that suggest the correct choice or at least eliminate some wrong ones. Below are some findings:

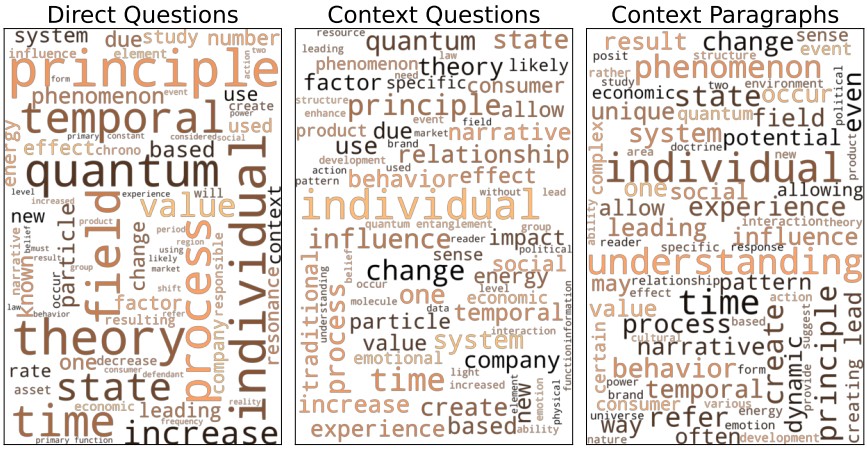

Figure 14: Word cloud for direct and context questions (including the four choices), and context paragraphs.

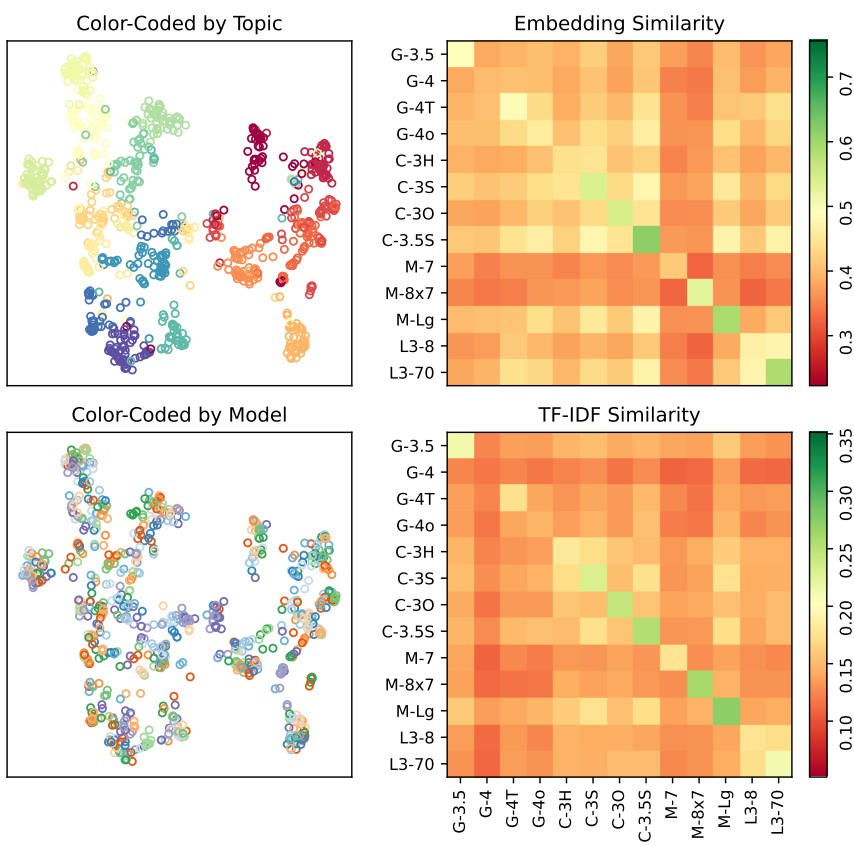

Figure 15: Visualization and similarity analyses for generated context paragraphs.

1. When the question introduces a term and only one or some choices mention this term, the correct answer is more likely to be among them. As an example, a question starts with "A company is experiencing a high rate of Flibulation", and only one choice mentions "Flibulation", which turns out to be the correct choice.

2. Related to the heuristic above, sometimes there are common sub-words between question statements and choice texts. For example, a question asks about "nucleocytokinesis", and one choice includes "cytoplasm" while others do not have any words that share sub-words with the concept.

3. The question describes a concept in a neutral to positive manner, but some choices sound slightly negative. They can often be eliminated. An example is CQ3 in Tab. 9, where the correct choice is less positive than the rest (note that the negation in the question).

4. There may be semantic matches between the question statement and choice texts. For example, the question asks "Which phenomenon is used to selectively reduce the strength of correlations in entangled quantum systems?" The answers are "A. Quantum superposition decay", "B. Quantum interference enhancement", "C. Quantum entanglement dampening", "D. Quantum entanglement creation". We can be pretty certain that the correct choice is between A and C.

It should be noted, however, that these identified heuristics are far from sufficient to achieve the observed model correctness rate, as shown in Fig. 16, which presents an expanded version of the correctness rate visualization in Fig. 4, with all answer models included. As we can see, models score very similarly to each other on each topic, but show variance across different topics.

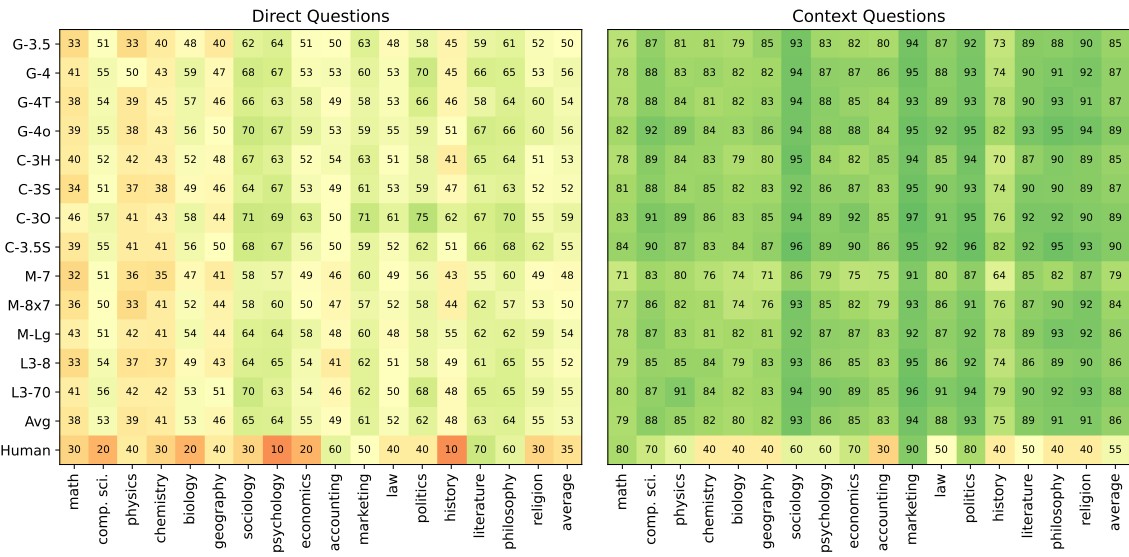

Figure 16: An expanded version of Fig. 4, showing the correctness rates of all AM models and their average.

# D Answer Shuffling Result Details

Fig. 17 reproduces an enlarged version of Fig. 7, with annotated cell values.

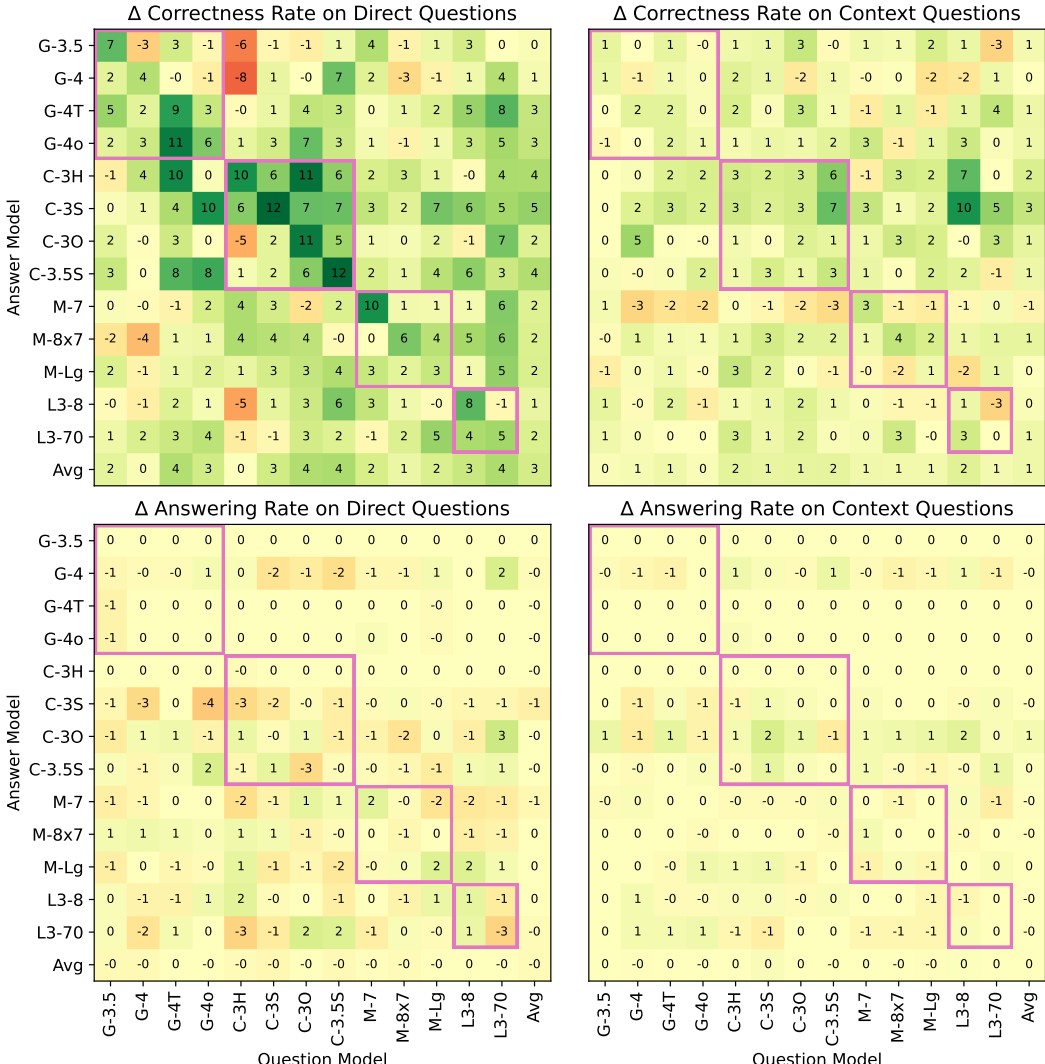

Figure 17: An enlarged version of Fig. 7, showing the effect of removing answer shuffling on correctness and answering rates. A positive (green) value means that the value would be higher without answer shuffling, and a negative (red) value means that the value would be lower.

# E    Fictionality Awareness Experiment Details

Fig. 18 shows the choice E selection rates on direct and question paragraphs, as well as fictionality detection rate for each (AM, QM) pair.

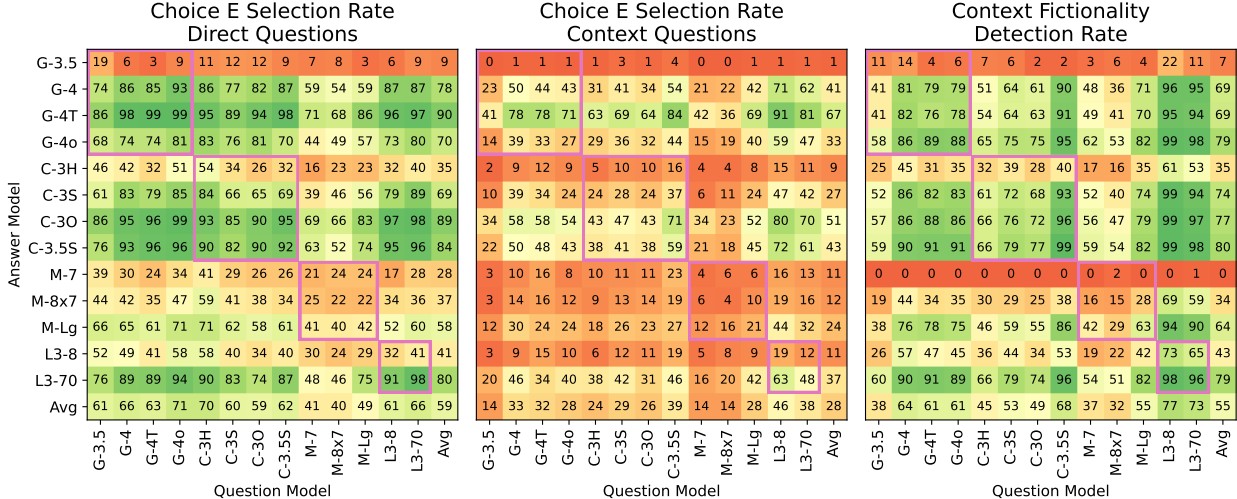

Figure 18: Choice E selection rate and fictionality detection rate for each (AM, QM) pair.

# F   Prompt for Sequential Question Generation

The prompt for generating five sequential questions is shown in Tab. 10.

| Role | Message |
|---|---|
| User | On the topic of physics, please write five multiple choice questions around concepts that are completely made up. Make sure that these questions are distinct from each other. Try to make each problem hard and challenging. In your question, do not say that the concept is hypothetical or fictional. Instead, treat it as if it were real and widely accepted. Use the following template:

Question 1: [question statement]

A. [choice A]
B. [choice B]
C. [choice C]
D. [choice D]

Answer: [the correct choice]

Question 2: [question statement]

A. [choice A]
B. [choice B]
C. [choice C]
D. [choice D]

Answer: [the correct choice]

Question 3: [question statement]

A. [choice A]
B. [choice B]
C. [choice C]
D. [choice D]

Answer: [the correct choice]

Question 4: [question statement]

A. [choice A]
B. [choice B]
C. [choice C]
D. [choice D]

Answer: [the correct choice]

Question 5: [question statement]

A. [choice A]
B. [choice B]
C. [choice C]
D. [choice D]

Answer: [the correct choice] |
| Model | *(the generated question and answer)* |

Table 10: The prompt for direct question generation. The underlined topic is replaced accordingly.

## G Universality Result Details

Fig. 19 presents the result of evaluating the correctness rates of other LLM types as AMs for all QMs, expanding on the average values across QMs shown in Fig. 11.

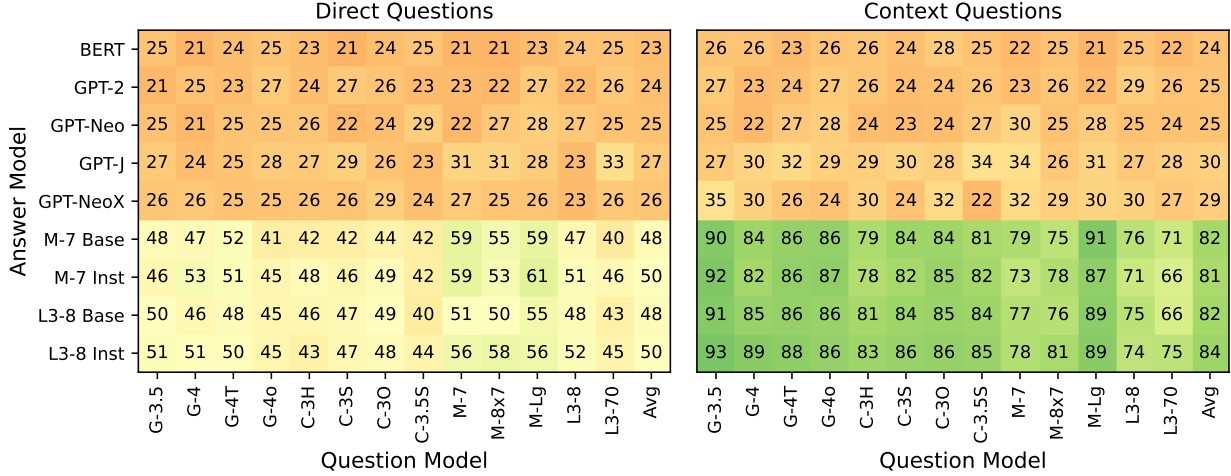

Figure 19: Correctness rates of other LLM types on all QMs.

## H Creative Writing Experiment Details

Tab. 11 and 12 present the prompts for generating direct and context questions in the creative writing experiment. The prompt used by the answer model is the same as before (Tab. 4). We generated 20 DQs and 20 CQs for each of the following 10 topics: friendship, family relationship, young adulthood, an ancient empire, an interpersonal conflict, a roadtrip, a childhood in poverty, future technology, a long-lasting war, an intergalactic civilization.

| Role | Message |
|---|---|
| User | Imagine a story about friendship with an intricate story plot. Without telling the story, write a question about one of its details, and also indicate the correct answer. Do not explicitly reference the story in the question (i.e., do not use phrases such as "in the story" or "according to the text"). Use the following template:

Question: [question statement]

A. [choice A]
B. [choice B]
C. [choice C]
D. [choice D]

Answer: [the correct choice] |
| Model | *(the generated question and answer)* |

Table 11: The prompt for direct question generation. The underlined topic is replaced accordingly.

| Role | Message |
|------|---------|
| User | You are an excellent writer. Write a short story of 3-5 paragraphs about friendship. Be creative, develop an intricate story plot and include lots of details. |
| Model | *(the generated concept and paragraph)* |
| User | Now, write a question about one of its details, and also indicate the correct answer. Do not explicitly reference the story in the question (i.e., do not use phrases such as "in the story" or "according to the text"). Use the following template:

Question: [question statement]

A. [choice A]
B. [choice B]
C. [choice C]
D. [choice D]

Answer: [the correct choice] |
| Model | *(the generated question and answer)* |

Table 12: The prompt for context-based question generation. The underlined topic is replaced accordingly.

Tab. 13 presents questions for the creative writing setup in Sec. 3.6, with correct choices in **bold**.

---

*DQ1: Mixtral 8x7B, Friendship*
Which character, before the climax, reveals their hidden identity to the protagonist?
A. The wise old mentor
B. **The long-lost sibling**
C. The jealous rival
D. The unsuspecting friend

---

*DQ2: GPT-4 Turbo, An ancient empire*
What was the name of the festival during which High Priest Zoroth attempted his betrayal?
A. Festival of the Harvest
B. Festival of the Sands
C. Festival of Lights
D. **Festival of Solaris**

---

*DQ3: GPT-4, An interpersonal conflict*
What was the real reason for Sophia secretly reading Lucy's diary?
A. Sophia was attempting to plagiarize Lucy's poems
B. Sophia was looking for incriminating evidence against Lucy
C. **Sophia was trying to understand Lucy's feelings towards her**
D. Sophia was snooping into Lucy's personal life out of curiosity

---

*DQ2: Mistral 7B, A childhood in poverty*
Which flower did Maria find and carefully nurture?
A. A red rose
B. A jasmine plant
C. A wilting daisy
D. **An abandoned sunflower seed**

---

*CQ2: Claude 3 Haiku, Future technology*
Which advanced technology is used by the protagonist to track the location of their missing loved one?
A. Quantum entanglement communication device
B. **Nanite swarm reconnaissance drones**
C. Gravitational warp field generator
D. Holographic surveillance system

---

*CQ3: Llama 3 70B, A long lasting war*
What is the primary reason for the disputed border between the Eastern Realm and the Western Empire?
A. **A powerful artifact was discovered in the contested territory.**
B. A centuries-old trade agreement was never formally ratified.
C. A long-forgotten treaty was intentionally mistranslated to conceal a hidden clause.
D. A series of brutal skirmishes sparked a cycle of revenge attacks.

---

Table 13: Example direct questions (DQs) and context questions (CQs), with correct choices in **bold**.

Fig. 20 shows the detailed breakdown of statistics presented in Tab. 8.

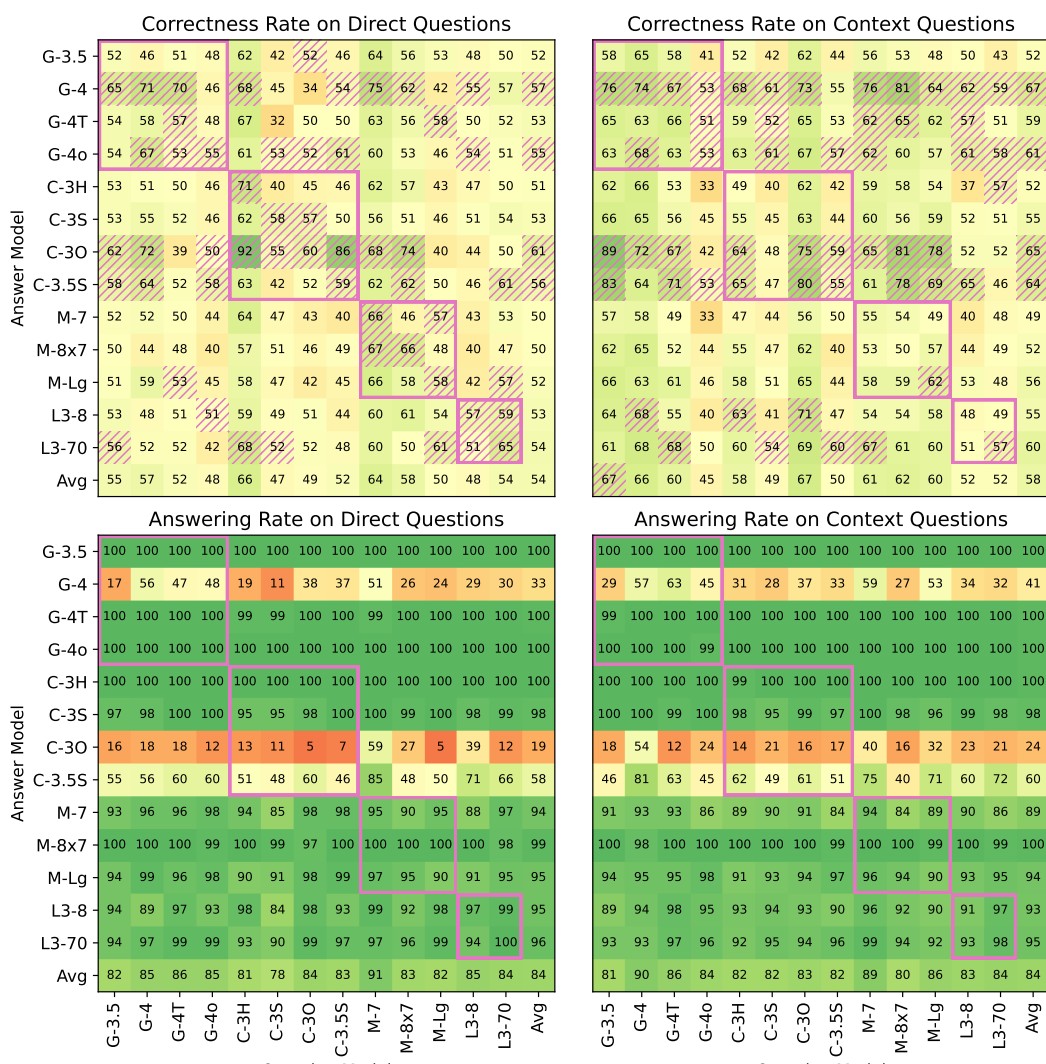

Figure 20: Correctness and answering rates for creative writing questions.

# I  Prompt Variation Experiment Details

Tab. 14 presents the exact prompts used in the prompt variation studies.

| Prompt | Text |
|---|---|
| 1 | Answer the following question. Be concise and give the answer only. *(the default prompt)* |
| 2 | Answer the following question with the letter for the best option. You do not need to explain. |
| 3 | Choose the best option for the question below. There is no need for any explanation. |
| 4 | Please select the best choice to answer the question, without giving any reasoning. |
| CoT | Answer the following question. First, use a paragraph to think step by step through the question. Then, output the final answer. |

Table 14: Four different prompt wordings (the first being the default one used in other experiments) and the zero-shot chain-of-thought prompt.

Fig. 21 lists the exact prompt template used by GPT-4o to grade models' free response answers.

You are an impartial teacher to grade the answer written by a student. Below, you are given a multiple-choice question and four choices, along with the correct choice. The student, however, answered the question based on the question statement only, without seeing the four choices. So he wrote his answer in a free text format. You need to score the quality of the student answer, on a scale of 0 to 100. Assign a score of 100 if the student answer is essentially the same as the correct choice, and a lower score accordingly for any deviation. Assign a score of 50 if the student answer is essentially the same as one of the wrong choices, and a lower score accordingly for any deviation. If the student answer is very different from any of the choices, assign a score close to 0. However, if the student refuses to answer the question by pointing out that the concept asked in the question appears to be fake or fictional, output "Refuse to answer" instead. Below is the question:

{Question}

A. {Choice A}
B. {Choice B}
C. {Choice C}
D. {Choice D}

The correct answer is: {Correct Choice Letter}. {Correct Choice Text}

The student answer is: {Answer Model Response}

Below, output an integer score for the student answer, or "Refuse to answer". Do not output anything else.

Figure 21: LLM-as-a-judge prompt template used to grade the free response answers.

Fig. 22 plot the detailed breakdown of correctness and answering rates under each of the four prompt wordings, grouped by the (AM, QM) pair. Except for some decrease on DQ answering rates under the first prompt (i.e., the default one used in other experiments), there are minimal effects of wording differences.

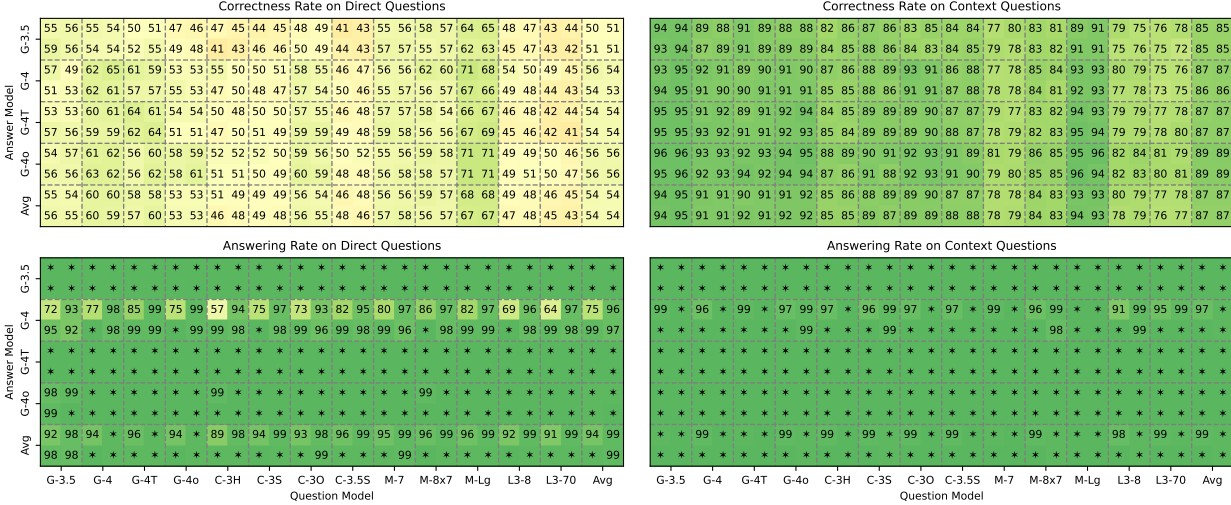

Figure 22: Correctness and answering rate breakdown for four different prompt wordings. Within each block of (AM, QM) pair, the four numbers represent those under the four wordings. Asterisks denote 100%.

Fig. 23 plot the detailed breakdown of correctness and answering rates under the chain-of-thought prompt. Fig. 24 plot the detailed breakdown of correctness scores and answering rates in the free-response answering setup (as judged by GPT-4o using the prompt in Fig. 21). The CoT result is very similar to the non-CoT ones, and while the correctness scores for the free-response setup is significantly lower, we see that most trends, notably the higher correctness scores and answering rates on CQs than DQs, are still preserved.

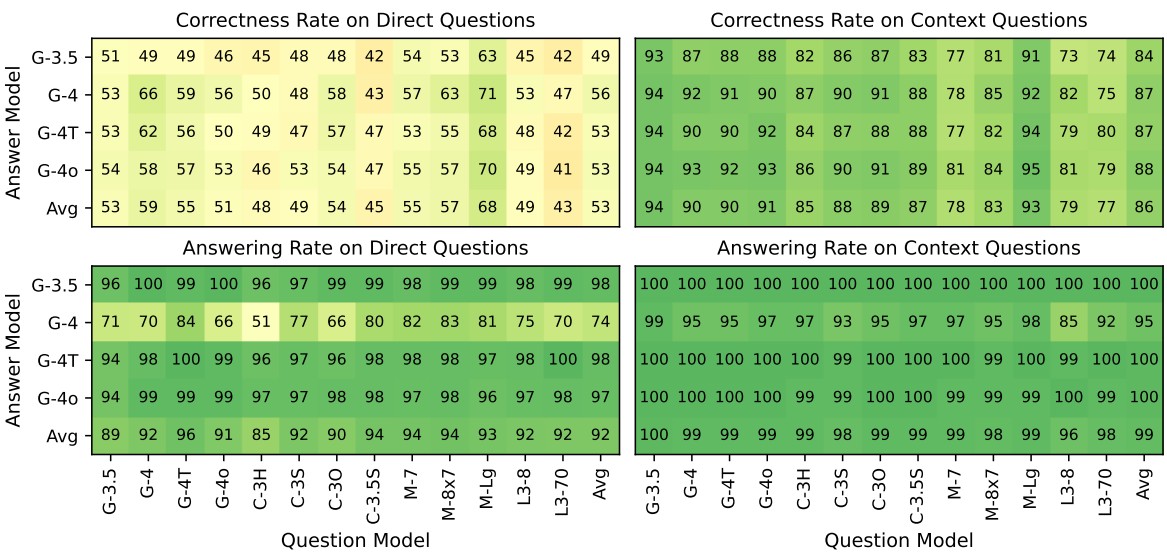

Figure 23: Correctness and answering rate breakdown for the chain-of-thought prompt.

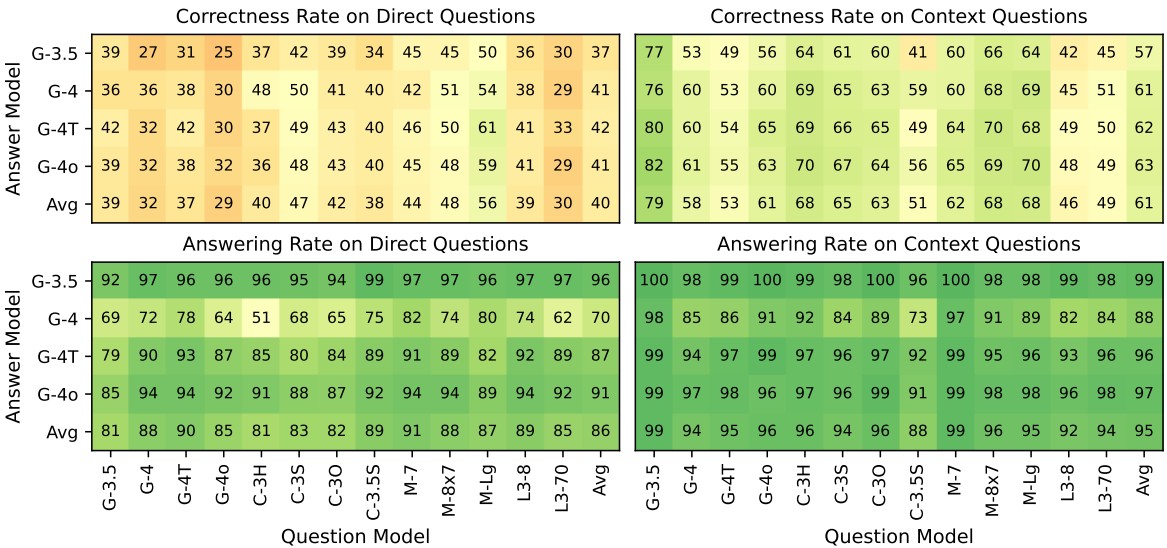

Figure 24: Correctness and answering rate breakdown for the chain-of-thought prompt.

