# OpenReview forum: "Shared Imagination: LLMs Hallucinate Alike"
_TMLR — Accepted by TMLR_

### Review · Reviewer_K71b · 2025-03-04

**Summary Of Contributions:**

The authors present a new study that addresses a very interesting question: what would LLM answer to questions about imaginative objects and scenarios? More concretely, the authors propose the imaginary question answering task where a model is prompted to generate multiple-choice questions about a given topic, and then ask another model to generate an answer. The authors propose this task in two settings: 1) where the model has to generate an answer based on just the question; 2) where the answer model has also access to additional context that can be used to provide an answer.

This authors complete this study using 14 different LLMs including both open-source and closed-source models. This provides a very comprehensive set of results for the evaluation that is used to shed light on 7 different interesting research questions such as different data characteristics, heuristics used by models to generate the correct choice, whether the models are actually able to spot imaginative contexts, and others. Overall, this represents a very interesting investigation in the overall internal representations that these models learn.

**Audience:**

Yes

**Claims And Evidence:**

Yes

**Requested Changes:**

1. Please clarify the criterion you used to select the different types of concepts you have used for the evaluation.
2. In paragraph 2.1, it would be useful to clarify whether you're following a likelihood-based evaluation or a generative one
3. I find the caption of Figure 1 a bit confusing because the authors refer to "bottom left" and "bottom right" but the image is on the same level (at least as far as I can tell).
4. Figure 6 at the moment it's supposed to show correctness rate for open-source models but there are also closed source ones. I think this is confusing because it's not possible to evaluate closed source models without access to likelihood scores.
5. Figure 9 refers to Figure 3 for the legend. This is a bit inconvenient and I would suggest the authors to report the legend in the same image.
6. Consider adding a final section where you discuss the results as a whole (think about it as a "Discussion" section) where you provide some indications regarding the implications of this work. For instance, it would be interesting to see how this work could be extend to multimodal models such as Vision and Language models or image/video generation models.

**Strengths And Weaknesses:**

## Strengths

1. Very interesting experimental setup that is automatically generated by different combinations of models using a pool of concepts
2. Seven different research questions to assess different capabilities and behaviours of the models
3. The authors test fourteen different models including both closed source and open source ones

## Weaknesses
1. Please report some additional details regarding the human evaluation. How did you set up this study? How many human annotators were involved? I think it's really important to have a robust human baseline for this study.
2. [minor] It would be interesting to see how Qwen-2 performs considering that it's been showing very good performance across the board
3. I find this work very interesting from a philosophical point of view (i.e., what's an imaginative world when everything is expressed via text and how do LLMs process it?) but I think the authors should spend some time in their manuscript to discuss the implications of this work for additional downstream applications and more fundamental research in the development of LLMs. I think this will make the manuscript even appealing to a wider audience.

---

> ### Author Response · Authors · 2025-04-18
>
> We thank the reviewer for your review, and are glad that you find our proposed IQA task interesting and our experiments systematic. We have updated the paper draft according to your requested changes and summarize our changes below:
>
> > RC1: Please clarify the criterion you used to select the different types of concepts you have used for the evaluation.
>
> We selected the 20 topics mainly inspired by the MMLU dataset, added to the 2nd paragraph of Sec. 2.2.
>
> > RC2: In paragraph 2.1, it would be useful to clarify whether you're following a likelihood-based evaluation or a generative one
>
> We use deterministic decoding and the answer choice letter (A, B, C, or D) is always a single token, so the two formulations are equivalent, which is also explained in the paper.
>
> > RC3: I find the caption of Figure 1 a bit confusing because the authors refer to "bottom left" and "bottom right" but the image is on the same level (at least as far as I can tell).
>
> Thank you for catching this. We have fixed it.
>
> > RC4: Figure 6 at the moment it's supposed to show correctness rate for open-source models but there are also closed source ones. I think this is confusing because it's not possible to evaluate closed source models without access to likelihood scores.
>
> Fig. 6 does study open-source models, which are arranged on the vertical axis as answer models. The horizontal axis lists the question models, which can be either open- or closed-source models. We have added the axis label to avoid any confusion.
>
> > RC5: Figure 9 refers to Figure 3 for the legend. This is a bit inconvenient and I would suggest the authors to report the legend in the same image.
>
> Thank you for your suggestion. We have added the color legend to Fig. 9.
>
> > RC6: Consider adding a final section where you discuss the results as a whole (think about it as a "Discussion" section) where you provide some indications regarding the implications of this work. For instance, it would be interesting to see how this work could be extend to multimodal models such as Vision and Language models or image/video generation models.
>
> We added a new Section 5 with various high-level discussions and have also expanded on the future work in Sec. 6 to be more substantive, including your suggestion of benchmarking multimodal models.
>
> Finally, for the first point of weakness on human evaluation, in the paper, this study is a relatively preliminary and exploratory one, carried out only with paper authors. We described the main findings in Appendix C. As most of the analyses, such as the warm-up effect, fictionality detection, etc. does not depend on the human guessing baseline, we decide to leave a more comprehensive human study to future work, with some ideas discussed in Sec. 6.
>
> Thank you again for your review. Please let us know if you would like to see further revisions.

---

### Review · Reviewer_1Uhb · 2025-03-24

**Summary Of Contributions:**

This paper proposes a new task, imaginary question answering (IQA), with its benchmarks. This new task aims to better model and measure the similarity among different large language models (LLMs). In that task, the authors propose to prompt one LLM to generate purely imaginary questions (e.g.,on completely made-up concepts in physics) and prompt another model to answer. They conduct a series of experiments and collect the phenomenon.

**Audience:**

Yes

**Claims And Evidence:**

Yes

**Requested Changes:**

see the weaknesses

**Strengths And Weaknesses:**

Strengths:

1.The proposed task is novel and new. The task sounds an interesting point to view some aspect of LLMs and it is also interesting to measure the similarity among different LLMs.

2.The paper is easy to follow and understand.

3.The proposed methods sound easy to implement and reproduce. The authors also release the source codes.

4.The experiments are solid and they authors collect a great number of  phenomenon under their proposed setting.

Weaknesses:

1.Although the proposed methods is easy and clear, the technoical novelty of the method is limited. Although this paper’s main contribution is around the proposed new task instead of the method, it would be better to xxx

2.  Some figures need to be refined. E.g. the figure for prompt text in section 3.5.

3.In the analysis section, the RQ should be a research question, which should not be a term (RQ1: Data Characteristics).

4.The related work does not provide a detailed description of the related research. It involves too much research topics and the connections between the topics are not so dense. For each research area, the author did not provide a comprehensive analyses of the related papers.

5.The authors provide many experimental results in multiple aspects but the insights and the in-depth analysis behind the results are not so sufficient. It is better the see some qualitative and generalized conclusion through the  experimental results. If there is no qualitative conclusion out-of-the-scope of the proposed task (IQA), the motivation of proposing such a task is not so strong.

---

> ### Author Response · Authors · 2025-04-18
>
> We thank the reviewer for your review, and are glad that you find our proposed IQA task novel and our experimental analysis solid and easy to reproduce. We have updated the paper draft according to your requested changes and summarize our changes below:
>
> > RC1: Although the proposed methods is easy and clear, the technoical novelty of the method is limited. Although this paper’s main contribution is around the proposed new task instead of the method, it would be better to xxx
>
> The main innovation of this paper is on the novelty of the IQA task setting, as the Strengths section of the review recognizes. Due to the lack of precedence in such investigation, we believe that it would be best to start from the more straightforward methods, analyzing model responses, perplexity, choice randomization ablation, human performance, etc. Given the already quite expansive extent of analyses, we defer novel methods to future work, where we give some ideas using mechanistic interpretability in the newly added Section 5. We would also appreciate any reviewer suggestions on this, as it seems like the last sentence of this point is left unfinished.
>
> > RC2: Some figures need to be refined. E.g. the figure for prompt text in section 3.5.
>
> We went over Sec. 3.5 but did not quite understand the suggestion by the reviewer. We would appreciate some further clarification.
>
> > RC3: In the analysis section, the RQ should be a research question, which should not be a term (RQ1: Data Characteristics).
>
> We changed the title to “Exploratory Data Analysis” for it to be more descriptive of the investigation for this RQ, but we would also be happy to take any reviewer suggestion here.
>
> > RC4: The related work does not provide a detailed description of the related research. It involves too much research topics and the connections between the topics are not so dense…
>
> Given the novelty and lack of precedence of this IQA task setup, we could not find any directly related prior work. Instead, our related work section focuses more on the broader phenomenon of model similarity and implications on model trustworthiness. Nonetheless, we expanded this section with more discussions and are happy to take reviewer suggestions for additional references.
>
> > RC5: The authors provide many experimental results in multiple aspects but the insights and the in-depth analysis behind the results are not so sufficient. It is better the see some qualitative and generalized conclusion through the experimental results…
>
> We added Sec. 5 on discussions of several high-level results and implications, which we believe would contextualize our work in the broader LLM research.
>
> Thank you again for your review. Please let us know if you would like to see further revisions.

---

### Review · Reviewer_RgkF · 2025-04-11

**Summary Of Contributions:**

- This paper introduces the Imaginary Question Answering (IQA) task, where one LLM generates imaginary multiple-choice questions on entirely fictional concepts, and another LLM attempts to answer these questions.

- The authors experiment with multiple recent LLMs (e.g., GPT series, Claude, Mistral, and Llama 3) and systematically prompt one model (the “question model”) to generate fictional multiple-choice questions, then ask another model (the “answer model”) to respond.

- Despite generating entirely fictional and thus logically unanswerable content, many models demonstrate a high degree of consistency and correctness when answering questions posed by other models. This phenomenon, referred to as "shared imagination" by authors, suggests significant underlying similarities among various LLMs.

- The work systematically explores this phenomenon across different settings, question generation modes, content types, context paragraphs, prompt lengths, sequential “warm-up” effects, and differences in model families.

- Their findings suggest that modern LLMs might share “imaginative” representations not strictly found in real data, raising questions regarding hallucination detection and the true creativity and independence of these models.

**Audience:**

Yes

**Broader Impact Concerns:**

Please see above.

**Claims And Evidence:**

Yes

**Requested Changes:**

I found the work quite interesting with thorough experimentation and engaging content. I would like to suggest a few improvements on certain aspects of the writing to significantly enhance the paper’s potential impact.

RC1. It is strongly recommended to expand explicitly on potential reasons why large language models (LLMs) frequently converge on similar fictional content in mechanical (or theoretical) aspects. Possible contributing factors to explore include overlaps in training datasets (this is mentioned in the paper but could be discussed more thoroughly), specific alignment or fine-tuning procedures, and inherent properties of autoregressive architectures. The authors could provide a detailed section or an extended paragraph to hypothesize how these factors might reinforce each other, resulting in a phenomenon akin to "shared imagination".

RC2. Clearly articulating how the proposed framework can be effectively utilized by researchers and practitioners would significantly improve the paper. Additionally, highlighting possible issues regarding user trust (particularly scenarios where the model’s fabricated outputs might mistakenly be perceived as factual) would strengthen the relevance of the discussion from an AI safety perspective. Explicitly addressing how incorrect agreement or shared illusions among models may undermine trustworthiness would also be valuable.

RC3. The paper touches upon crucial ethical considerations regarding misinformation and the trustworthiness of language models. To enhance this important discussion, the authors should systematically address potential risks associated with generating convincing yet completely fabricated content, outlining clearly the dangers of misuse. Additionally, greater clarity on recommended mitigation strategies and ethical guidelines for both developers and end-users of LLM technologies would significantly reinforce the practical utility and impact of this work.

**Strengths And Weaknesses:**

S1. The paper introduces a creative concept IQA, which provides a new lens for understanding LLM behavior. Compared to standard factual QA, IQA spotlights how models handle content that is intentionally fictional or impossible to verify by external references. Findings are interesting and could reshape how hallucination and creativity in LLMs are understood.

S2. Methodology is clear and robust, systematically exploring numerous dimensions such as question types (direct and context-based), model types, and various experimental setups.

S3. The analysis is thorough. The authors test multiple models (GPT-3.5, GPT-4, Claude variants, Mistral, Llama 3) and systematically vary prompts, question-generation procedures, and topics, and delve deeply into key dimensions (e.g., impact of question order, question length, base vs. instruction-tuned models, and variations in prompt structure). It explores multiple plausible heuristics and explanations for observed phenomena.

---

W1. The mechanisms driving the "shared imagination" phenomenon are not deeply explained; the theoretical underpinnings remain somewhat speculative. While the authors thoroughly document the shared imagination phenomenon, the manuscript stops short of providing an in-depth mechanistic explanation of why these models converge on similar fictitious content. Including at least a speculative discussion on possible reasons (e.g., overlap in training data or alignment processes) would strengthen the paper.

W2. The manuscript uses external embeddings to cluster or visualize the imaginary questions. This is a minor critique, but the choice itself might introduce an external bias. While those results are indicative, an ablation or sanity check with alternative embeddings could add robustness.

---

> ### Author Response · Authors · 2025-04-18
>
> We thank the reviewer for your review, and are glad that you find our proposed IQA task novel and our experimental analysis robust and systematic. We have updated the paper draft according to your requested changes and summarize our changes below:
>
> > RC1: It is strongly recommended to expand explicitly on potential reasons why large language models (LLMs) frequently converge on similar fictional content in mechanical (or theoretical) aspects…
>
> We dedicated a new Section 5 on more high-level discussions, including the role of training data and potential ideas of using mechanistic interpretability to further study the phenomenon. We have also expanded on the future work discussion of Section 6 to present some more concrete plans that broaden the analysis.
>
> > RC2 and RC3: Clearly articulating how the proposed framework can be effectively utilized by researchers and practitioners would significantly improve the paper. Additionally, highlighting possible issues regarding user trust… To enhance this important discussion, the authors should systematically address potential risks associated with generating convincing yet completely fabricated content…
>
> We added more detailed discussion in Section 5 on broader implications on LLM trustworthiness.
>
> Thank you again for your review. Please let us know if you would like to see further revisions.

---

### Author Response · Authors · 2025-04-18

Dear Reviewers and Area Chair,

Thank you for your review of this submission. We have responsed to each review and updated the paper draft. The biggest change is the addition of Section 5, where we go in depth on discussing some overarching ideas and findings of the experiments. In addition, we also expanded on Section 6 and made the future work discussions more substantive. Last, we made several changes on various details requested by reviewers. Please let us know if you would like to see any further revisions!

Best,

Paper Authors

---

### Decision · Action_Editor_YMqD · 2025-06-25

**Recommendation:** Accept as is

**Additional Comments:**

Some reviewers make some important comments for further revision.
Although ACs' decision is "Accept as it", they may need to addressed for the camera ready version.

**Audience:**

Yes

**Audience Explanation:**

All authors agree that the problem of this submission is novel and interesting.

**Claims And Evidence:**

Yes

**Claims Explanation:**

After the revision, all reviewers converge to the accept opinion and show their satisfaction to the revision.